# Private Federated Frequency Estimation: Adapting to the Hardness of the Instance

**Jingfeng Wu**[*]
Johns Hopkins University
uuujf@jhu.edu

**Wennan Zhu**
Google Research
wennanzhu@google.com

**Peter Kairouz**
Google Research
kairouz@google.com

**Vladimir Braverman**
Rice University
vb21@rice.edu

## Abstract

In *federated frequency estimation* (FFE), multiple clients work together to estimate the frequencies of their collective data by communicating with a server that respects the privacy constraints of Secure Summation (SecSum), a cryptographic multi-party computation protocol that ensures that the server can only access the sum of client-held vectors. For single-round FFE, it is known that *count sketching* is nearly information-theoretically optimal for achieving the fundamental accuracy-communication trade-offs [Chen et al., 2022]. However, we show that under the more practical multi-round FEE setting, simple adaptations of count sketching are strictly sub-optimal, and we propose a novel hybrid sketching algorithm that is *provably* more accurate. We also address the following fundamental question: *how should a practitioner set the sketch size in a way that adapts to the hardness of the underlying problem*? We propose a two-phase approach that allows for the use of a smaller sketch size for simpler problems (e.g. near-sparse or light-tailed distributions). We conclude our work by showing how differential privacy can be added to our algorithm and verifying its superior performance through extensive experiments conducted on large-scale datasets.

## 1 Introduction

In many distributed learning applications, a server seeks to compute population information about data that is distributed across multiple clients (users). For example, consider a distributed frequency estimation problem where there are $n$ clients, each holding a local data from a domain of size $d$, and a server that aims to estimate the frequency of the items from the $n$ clients with the minimum communication cost. This task can be done efficiently by letting each client *binary encode* their data and send the encoding to the server, at a local communication bandwidth cost of $\log(d)$ bits. With the binary encoding, the server can faithfully decode *each* local data and compute the global frequency vector (i.e., the normalized histogram vector).

However, the local data could be sensitive or private, and the clients may wish to keep it hidden from the server. The above binary encoding communication method, unfortunately, allows the server to observe each individual local data, and therefore may not satisfy the users' privacy concerns. *Federated Analytics* (FA) [Ramage and Mazzocchi, 2020, Zhu et al., 2020] addresses this issue by developing new methods that enable the server to learn population information about the clients while preventing the server from prying on any individual local data. In particular, a cryptographic

---

[*]Work done during an internship at Google Research

multi-party computation protocol, *Secure Summation* (`SecSum`) [Bonawitz et al., 2016], has become a widely adopted solution to provide data minimization guarantees for FA [Bonawitz et al., 2021]. Specifically, `SecSum` sets up a communication protocol between clients and the server, which injects carefully designed additive noise to each data that cancels out when *all of the local data are summed together*, but blurs out (information theoretically) each individual local data. Under `SecSum`, the server is able to faithfully obtain the correct summation of the data from all clients but is unable to read a single local data. The *federated frequency estimation* (FFE) problem refers to the distributed frequency estimation problem under the constraint of `SecSum`. Clearly, the binary encoding method is not compatible with `SecSum`, because when the binary encoding is passed to the server through `SecSum`, the server only gets the summation of the binary encodings of the users' data, which does not provide sufficient information for computing the global frequency vector.

A naive approach to FFE can be accomplished by employing *one-hot encoding*: each client encodes its local data into a $d$-dimensional one-hot vector that represents the local frequency vector and sends it to the server through `SecSum`. Then the server observes the summation of the local frequency vectors using `SecSum` and scales it by the number of clients to obtain the true frequency vector. However, this one-hot encoding approach costs $\Theta(d \log(n))$ bits of communication per client. This is because `SecSum` adds noise from a finite group of size $\Theta(\log n)$ to each component of the $d$-dimensional local frequency vector [Bonawitz et al., 2016] to avoid overflows. With a linear dependence on domain size $d$, the one-hot encoding approach is inefficient for large domain problems, which is a common setting in practice. In what follows, we focus on the regime where $d > n$.

Recently, linear compression methods were applied to mitigate the high communication cost issue for FFE with large domains [Chen et al., 2021, 2022]. The idea is to first *linearly compress* the local frequency vector into a lower dimensional vector before sending it to the server through `SecSum`; as linear compression operators commute with the summation operator, the server equivalently observes a linearly compressed global frequency vector though `SecSum` (after rescaling by the number of clients). The server then applies standard decoding methods to approximately recover the global frequency vector from the linearly compressed one. In particular, Chen et al. [2022] show that `CountSketch` [Charikar et al., 2002] (among other sparse recovery methods) can be used as a linear compressor for the above purpose, which leads to a communication bandwidth cost of $\mathcal{O}(n \log(d) \log(n))$ bits. Therefore when $d > n$, `CountSketch` achieves a saving in local communication bandwidth compared to the one-hot encoding method that requires $\Theta(d \log(n))$ bits. Moreover, Chen et al. [2022] show that for FFE with a single communication round, an $\Omega(n \log(d))$ local communication cost is information-theoretically *unavoidable* for worst-case data distributions, i.e., we cannot do better without making additional assumptions on the global frequency vector.

**Contributions.**    In this work, we make three notable extensions to `CountSketch` for FFE problems.

1. We show that the way Chen et al. [2022] set up the sketch size (linear in the number of clients $n$) is often *pessimistic* (see Corollary 2.4). In fact, in the streaming literature, the estimation error induced by `CountSketch` is known to adapt to the tail norm of the global frequency vector [Minton and Price, 2014], which is often sub-linear in $n$. Motivated by this, we provide an easy-to-use, two-phase approach that allows practitioners to determine the necessary sketch size by automatically adapting to the hardness of the FFE problem instance.

2. We consider FFE with multiple communication rounds, which better models practical deployments of FA where aggregating over (hundreds of) millions of clients in a single round is not possible due to device availability and limited server bandwidth. We propose a new multi-round sketch algorithm called `HybridSketch` that *provably* performs better than simple adaptations of `CountSketch` in the multi-round setting, leading to further improvements in the communication cost. Surprisingly, we show that `HybridSketch` adapts to the tail norm of a *heterogeneity vector* (see Theorem 3.2). Moreover, the tail of the heterogeneity vector is always no heavier, and could be much lighter, than that of the global frequency vector, explaining the advantage of `HybridSketch`. For instance, on the C4 dataset [Bowman et al., 2020] with a domain size of $d = 150,868$ and $150,000$ users, we show that our method can reduce the sketch size by $83\%$ relative to simple sketching methods when the number of sketch rows is not very large.

3. We extend the Gaussian mechanism for `CountSketch` proposed by Pagh and Thorup [2022], Zhao et al. [2022] to the multi-round FFE setting to show how our sketching methods can be made differentially private [Dwork et al., 2006]. We also characterize the trade-offs between accuracy and privacy for our proposed method.

We conclude by verifying the performance of our methods through experiments conducted on several large-scale datasets. All proofs and additional experimental results are differed to the appendices.

## 2 Adapting `CountSketch` to the Hardness of the Instance

In this part, we focus on single-round FFE and show how `CountSketch` can achieve better results when the underlying problem is simpler. Motivated by this, we also provide a two-phase method for auto-tuning the hyperparameters of `CountSketch`, allowing it to automatically adapt to the hardness of the instance.

**Single-Round FFE.** Consider $n$ clients, each holding an item from a discrete domain of size $d$. The items are denoted by $x_t \in [d]$ for $t = 1, \ldots, n$. Then the frequency of item $j$ is denoted by

$$f_j := \frac{1}{n} \sum_{t=1}^n \mathbb{1}\left[x_t = j\right].$$

We use $\mathbf{x}_t$ to denote the one-hot representation of $x_t$, i.e., $\mathbf{x}_t = \mathbf{e}_{x_t}$ where $(\mathbf{e}_t)_{t=1}^d$ refers to the canonical basis. Then the frequency vector can be denoted by

$$\mathbf{f} := (f_1, \ldots, f_d)^\top = \frac{1}{n} \sum_{t=1}^n \mathbf{x}_t \in [0,1]^d.$$

In single-round FFE, the $n$ clients communicate with a server once under the constraint of `SecSum`, and aim to estimate the frequency vector $\mathbf{f}$. Note that `SecSum` ensures that the server can only observe the sum of the local data.

---

**Algorithm 1** COUNT SKETCH FOR FEDERATED FREQUENCY ESTIMATION

---

**Require:** $n$ clients with local data $x_t \in [d]$ for $t = 1, \ldots, n$. Sketch length $L$ and width $W$.
1: The server prepares independent hash functions and broadcasts them to each client:

$$h_\ell : [d] \to [W], \ \sigma_\ell : [d] \to \{\pm 1\}, \text{ for } \ell \in [L].$$

2: **for** Client $t = 1, \ldots, n$ in parallel **do**
3:     Client $t$ encodes the local data $x_t \in [d]$ to $\texttt{enc}(x_t) \in \mathbb{R}^{L \times W}$ where

$$\left(\texttt{enc}(x_t)\right)_{\ell,k} = \mathbb{1}\left[h_\ell(x_t) = k\right] \cdot \sigma_\ell(x_t) \text{ for } \ell \in [L], \ k \in [W].$$

4:     Client $t$ sends $\texttt{enc}(x_t) \in \mathbb{R}^{L \times W}$ to `SecSum`.
5: **end for**
6: `SecSum` receives $\left(\texttt{enc}(x_t)\right)_{t=1}^n$ and only reveals the summation $\sum_{t=1}^n \texttt{enc}(x_t)$ to the server.
7: **for** Item $j = 1, \ldots, d$ in parallel **do**
8:     Server produces $L$ estimators for $f_j$:

$$\texttt{dec}(j; \ell) := \sigma_\ell(j) \cdot \left(\frac{1}{n} \sum_{t=1}^n \texttt{enc}(x_t)\right)_{\ell, h_\ell(j)} \text{ for } \ell \in [L].$$

9:     Server computes the median of the $L$ estimators:

$$\texttt{dec}(j) := \texttt{median}\{\texttt{dec}(j; \ell) : \ \ell \in [L]\}.$$

10: **end for**
11: **return** $(\texttt{dec}(j))_{j=1}^d$ as estimate to $(f_j)_{j=1}^d$.

---

**Count Sketch.** `CountSketch` is a classic streaming algorithm that dates back to [Charikar et al., 2002]. In the literature of streaming algorithms, `CountSketch` has been extensively studied and is known to be able to adapt to the hardness of the problem instance. Specifically, `CountSketch` of a fixed size induces an estimation error adapting to the tail norm of the global frequency vector [Minton and Price, 2014].

A recent work by Chen et al. [2022] apply `CountSketch` to single-round FFE. See Algorithm 1 for details. They show that `CountSketch` approximately solves single-round FFE with a communication cost of $\mathcal{O}(n \log(d) \log(n))$ bits per client. Moreover, they show $\Omega(n \log(d))$ bits of communication per client is unavoidable for worst-case data distributions (unless additional assumptions are made),

confirming its near optimality. However, the results by Chen et al. [2022] are *pessimistic* as they ignore the ability of `CountSketch` to adapt to the hardness of the problem instance. In what follows, we show how the performance of `CountSketch` can be improved when the underlying problem becomes simpler.

We first present a problem-dependent accuracy guarantee for `CountSketch` of a fixed size, $L \times W$, that gives the sharpest bound to our knowledge. The bound is due to Minton and Price [2014] and is restated for our purpose.

**Proposition 2.1** (Restated Theorem 4.1 in Minton and Price [2014]). *Let* $(\hat{f}_j)_{j=1}^d$ *be estimates produced by* `CountSketch` *(see Algorithm 1). Then for each* $p \in (0, 1)$, $W \geq 2$ *and* $L \geq \log(1/p)$, *it holds that: for each* $j \in [d]$, *with probability at least* $1 - p$,

$$|\hat{f}_j - f_j| < C \cdot \sqrt{\frac{\log(1/p)}{L} \cdot \frac{1}{W} \cdot \sum_{i>W} (f_i^*)^2},$$

*where* $(f_i^*)_{i \geq 1}$ *refers to* $(f_i)_{i \geq 1}$ *sorted in non-increasing order, and* $C > 0$ *is an absolute constant.*

For the concreteness of discussion, we will focus on $\ell_\infty$ as a measure of estimation error in the remainder of the paper. Our discussions can be easily extended to $\ell_2$ or other types of error measures. Proposition 2.1 directly implies the following $\ell_\infty$-error bounds for `CountSketch` (by an application of union bound).

**Corollary 2.2** ($\ell_\infty$-error bounds for `CountSketch`). *Consider Algorithm 1. Then for each* $p \in (0, 1)$, $L = \log(d/p)$ *and* $W \geq 2$, *it holds that: with probability at least* $1 - p$,

$$\|\texttt{dec}(\cdot) - \mathbf{f}\|_\infty < C \cdot \sqrt{\frac{1}{W} \cdot \sum_{i>W} (f_i^*)^2}, \tag{1}$$

*where* $C > 0$ *is an absolute constant. In particular,* (1) *implies that*

$$\|\texttt{dec}(\cdot) - \mathbf{f}\|_\infty < C/W.$$

According to Corollary 2.2, the estimation error is smaller when the underlying frequency vector $(f_i^*)_{i \geq 1}$ has a lighter tail. In other words, `CountSketch` requires a smaller communication bandwidth when the global frequency vector has a lighter tail. Our next Corollary 2.3 precisely characterizes this adaptive property in terms of the required communication bandwidth. To show this, we will need the following definition on the *probable approximate correctness* of an estimate.

**Definition 1** (($\tau, p$)-correctness). *An estimate* $\hat{\mathbf{f}} := (\hat{f}_i)_{i=1}^d$ *of the global frequency vector* $\mathbf{f} := (f_i)_{i=1}^d$ *is* ($\tau, p$)-*correct if*

$$\mathbb{P}\left\{ \|\hat{\mathbf{f}} - \mathbf{f}\|_\infty := \max_i |\hat{f}_i - f_i| > \tau \right\} < p.$$

**Corollary 2.3** (Oracle sketch size). *Fix parameters* $\tau, p \in (0, 1)$. *Then for* `CountSketch` *(see Algorithm 1) to produce an* ($\tau, p$)-*correct estimate, it suffices to set the sketch size to* $L = \log(d/p)$ *and*

$$W = C \cdot \min\left\{ \left( \#\{f_i : f_i \geq \tau\} + \tfrac{1}{\tau^2} \cdot \sum_{f_i < \tau} f_i^2 \right), \; n \right\}, \tag{2}$$

*where* $C > 0$ *is an absolute constant. In particular, the width* $W$ *in* (2) *satisfies*

$$W \leq W_{worst} := C \cdot \min\left\{ 2/\tau, \; n \right\}. \tag{3}$$

Corollary 2.3 suggests that the sketch size can be set smaller if the underlying frequency vector has a lighter tail. When translated to the communication bits per client (that is $\mathcal{O}(L \cdot W \cdot \log(n))$, where $\log(n)$ accounts for the cost of `SecSum`), Corollary 2.3 implies that `CountSketch` requires

$$\mathcal{O}\left( \min\left\{ \#\{f_i \geq \tau\} + \tfrac{1}{\tau^2} \sum_{f_i < \tau} f_i^2, \; n \right\} \log(d) \log(n) \right) \leq \mathcal{O}(\min\{1/\tau, n\} \log(d) \log(n)) \tag{4}$$

bits of communication per client to be ($\tau, p$)-correct. In the worst case where $(f_i)_{i=1}^d$ is $\Theta(n)$-sparse and $\tau = \mathcal{O}(1/n)$, (4) nearly matches the $\Omega(n \log(d))$ information-theoretic worst-case communication cost shown in Chen et al. [2022], ignoring the $\log(n)$ factor from `SecSum`. However, in practice, $(f_i)_{i=1}^d$ has a fast-decaying tail, and (4) suggests that `CountSketch` can use less communication to solve the problem. We provide the following examples for a better illustration of the sharp contrast between the worst and typical cases.

**Corollary 2.4** (Examples). *Fix parameters $\tau, p \in (0,1)$. Consider Algorithm 1 with sketch length $L = \log(d/p)$. Then in each case for Algorithm 1 to produce an $(\tau, p)$-correct estimate for $\tau > 1/n$:*

1. *When $f_i \propto 2^{-i}$, it suffices to set $W = \Theta(\log(1/\tau))$.*
2. *When $f_i \propto i^{-a}$ for $a > 1$, it suffices to set $W = \Theta(\tau^{-1/a})$.*
3. *When $f_i \propto i^{-1} \log^{-b}(i)$ for $b > 1$, it suffices to set $W = \Theta(\tau^{-1} \log^{-b}(1/\tau))$.*
4. *When $f_i = 10/n$ for $i = 1, \ldots, n/10$, it suffices to set $W = \Theta(1/\tau)$.*

**A Two-Phase Method for Hyperparameter Setup.** Corollary 2.3 allows to use `CountSketch` with a smaller width for an easier single-round FFE problem, saving communication bandwidth. However, the sketch size formula given by (2) in Corollary 2.3 relies on crucial information of the frequency $(f_i)_{i \geq 1}$, i.e., $\#\{f_i : f_i \geq \tau\}$ and $\sum_{f_i < \tau} f_i^2$, which are unknown to who sets the sketch size. Thus, it is unclear if and how these gains can be realized in practical deployments.

We resolve this quandary by observing that in practice, the frequency vector often follows Zipf's law [Cevher, 2009, Powers, 1998]. This motives us to conservatively model the global frequency vector by a polynomial with parameters. By doing so, we can first run a small `CountSketch` to collect data from a (randomly sampled) fraction of the clients for estimating the parameters. Then based on the estimated parameter, we can set up an appropriate sketch size for a `CountSketch` to solve the FFE problem. This two-phase method is formally stated as follows.

We approximate the (sorted) global frequency vector $(f_i^*)_{i=1}^d$ by a polynomial [Cevher, 2009] with two parameters $\alpha > 0$ and $\beta > 0$, such that

$$f_i^* \approx \mathrm{poly}\,(i; \alpha, \beta), \quad \mathrm{poly}\,(i; \alpha, \beta) := \begin{cases} \beta \cdot i^{-\alpha}, & i \leq i^*; \\ 0, & i > i^*, \end{cases}$$

where $i^* := \max\{i : \sum_{j=1}^i \beta \cdot j^{-\alpha} \leq 1\}$ is set such that $\mathrm{poly}\,(i; \alpha, \beta)$ is a valid frequency vector. Here's an executive summary of the proposed approach for setting the sketch size.

1. Randomly select a subset of clients (e.g., $5,000$ out of $10^6$.)
2. Fix a small sketch (e.g., $16 \times 100$) and run Algorithm 1 with the subset of clients to obtain an estimate $(\tilde{f}_i)$.
3. Use the top-$k$ values (e.g., top 20) from $\tilde{f}_i$ to fit a polynomial with parameter $\alpha$ and $\beta$ (under squared error).
4. Solve Equation (4) under the approximation that $f_i^* \approx \beta \cdot i^\alpha$ and output $W$ according to the result.

**Experiments.** We conduct three sets of experiments to verify our methods. In the first set of experiments, we simulate a single-round FFE problem with the Gowalla dataset [Cho et al., 2011]. The dataset contains $6,442,892$ lists of location information. We first construct a domain of size $d = 175,000$, which corresponds to a grid over the US map. Then we sample $n = d/10 = 17,500$ lists of the location information (that all belong to the domain created) to represent the data of $n$ clients, uniformly at random. This way, we set up a single-round FFE problem with $n = 17,500$ clients in a domain of size $d = 175,000$. In the experiments, we fix the confidence parameter to be $p = 0.1$ and the sketch length to be $L = \ln(2d/p) \approx 16$. The targeted $\ell_\infty$-error $\tau$ is chosen evenly from $(10^{-3}, 10^{-1})$. We only test $\tau > 20/n$ because it is less important to estimate frequencies over items with small counts (say, 20). For `CountSketch`, we compute sketch width with three strategies, using (2) (called "instance optimal"), using (3) (called "minimax optimal"), and using the two-phase method. We emphasize that the "instance optimal" method is not a practical algorithm as it requires access to unknown information about the frequency; we use it only for demonstrating the correctness of our theory. We set all constant factors to be 2. The results are presented in Figures 1(a) and (b). We observe that the "minimax optimal" way of hyperparameter choice is in fact suboptimal in practice, and is improved by the "instance optimal" and the two-phase strategies.

In the second set of experiments, we run simulations on the "Colossal Clean Crawled Corpus" (C4) dataset [Bowman et al., 2020], which consists of clean English text scraped from the web. We treat each domain in the dataset as a user and calculate the number of examples each user has. The domain size $d = 150,868$, which is the maximum example count per user. We randomly sample $n = 150,000$ users from the dataset. We fix the sketch length to be $L = 5$. Other parameters are the same as the Gowalla dataset. The results are presented in Figures 1(d) and (e), and are consistent with what we have observed in the Gowalla simulations.

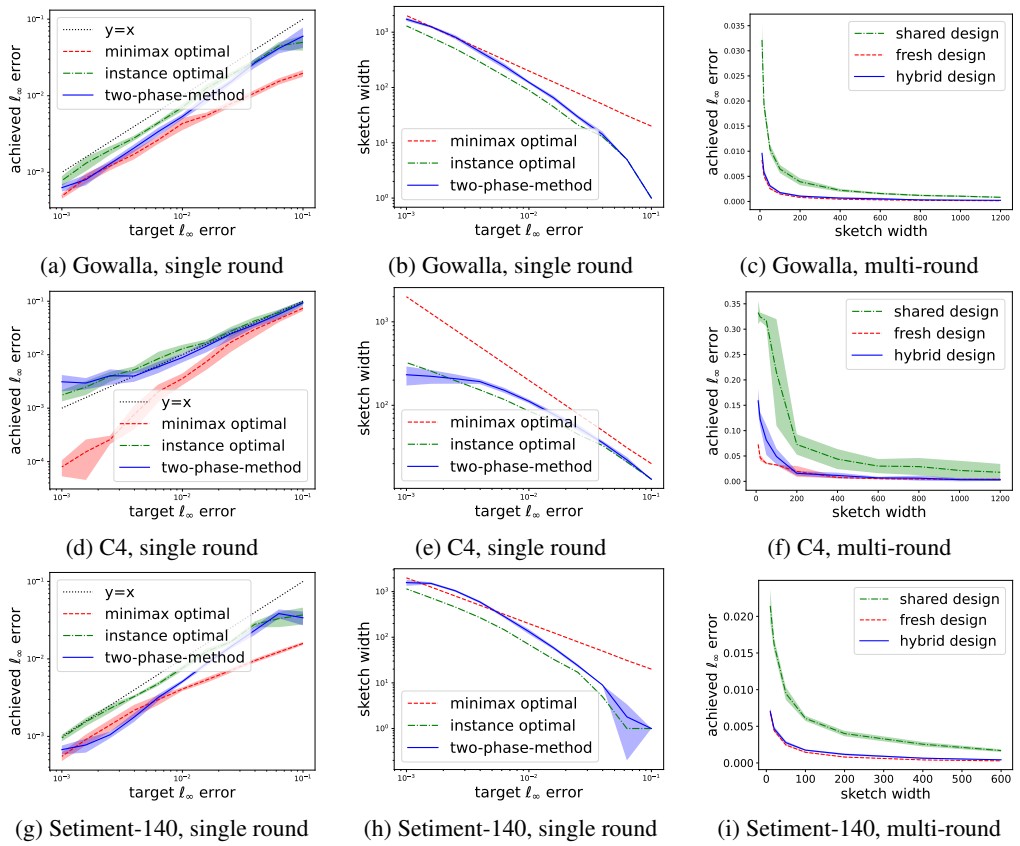

(a) Gowalla, single round      (b) Gowalla, single round      (c) Gowalla, multi-round

(d) C4, single round      (e) C4, single round      (f) C4, multi-round

(g) Setiment-140, single round      (h) Setiment-140, single round      (i) Setiment-140, multi-round

Figure 1: Single-round and multi-round FFE simulations. Subfigures (a) and (b) compare different hyperparameter strategies for `CountSketch` in a single-round FFE problem on the Gowalla dataset [Cho et al., 2011]. Subfigure (c) compares three sketch methods in a multi-round FFE problem on the Gowalla dataset. Subfigures (d), (e), and (f) are counterparts of subfigures (a), (b), and (c), respectively, but on the C4 [Bowman et al., 2020] dataset. Similarly, subfigures (g), (h), and (i) are counterparts of subfigures (a), (b), and (c), respectively, but on the Sentiment-140 [Go et al., 2009] dataset.

In the third set of experiments, we run simulations on a Twitter dataset Sentiment-140 [Go et al., 2009]. The dataset contains $d = 739,972$ unique words from $N = 659,497$ users. We randomly sample one word from each user to construct our experiment dataset. The number of rounds $M = 10$, and in each round, $n = N/10 = 65,949$ clients participate. The algorithm setup is the same as in the Gowalla experiments. Results are provided in Figures 1(g) and (h), and are consistent with our prior understandings.

## 3 Sketch Methods for Multi-Round Federated Frequency Estimation

In practice, having all clients participate in a single communication round is usually infeasible due to the large number of devices, their unpredictable availability, and limited server bandwidth [Bonawitz et al., 2019]. This motivates us to consider a multi-round FFE setting.

**Multi-Round FFE.** Consider a FFE problem with $M$ rounds of communication. In each round, $n$ clients participate, each holding an item from a universe of size $d$. The items are denoted by $x_t^{(m)} \in [d]$, where $t \in [n]$ denotes the client index and $m \in [M]$ denotes the round index. For simplicity, we assume in each round a new set of clients participate. So in total there are $N = Mn$ clients. Then the frequency of item $j$ is now denoted by

$$f_j := \frac{1}{Mn} \sum_{m=1}^{M} \sum_{t=1}^{n} \mathbb{1}\left[x_t^{(m)} = j\right].$$

For the $m$-th round, the local frequency is denoted by $f_j^{(m)} := \frac{1}{n} \sum_{t=1}^{n} \mathbb{1}\left[x_t^{(m)} = j\right]$. Clearly, we have $f_j = \frac{1}{M} \sum_{m=1}^{M} f_j^{(m)}$. Similarly, we use $\mathbf{x}_t^{(m)}$ to denote the one-hot representation of $x_t^{(m)}$, i.e.,

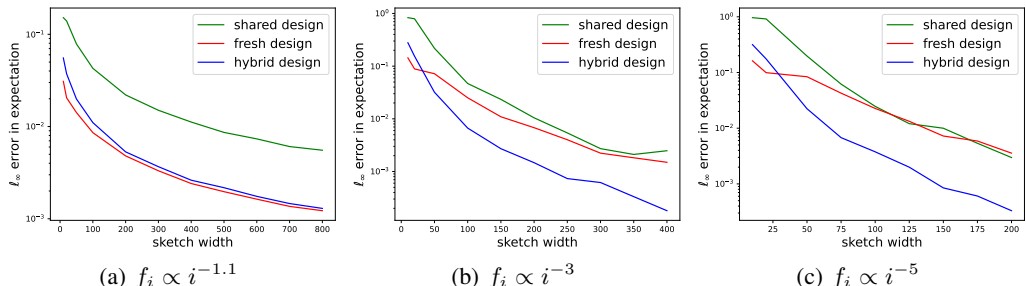

$$(a)\ f_i \propto i^{-1.1} \qquad (b)\ f_i \propto i^{-3} \qquad (c)\ f_i \propto i^{-5}$$

Figure 2: Shared vs. Hybrid vs. Fresh Sketches. We refer the reader to Section 3 for the definitions of the three methods. We compute the expected $\ell_\infty$-error for shared/hybrid/fresh sketches for a homogeneous, multi-round FFE problem. The domain size is $d = 10^5$. The number of rounds is $M = 10$. In all setups, the sketch length is fixed to $L = 5$. In every setting, the $\ell_\infty$ error is averaged with $1,000$ random repeats for simulating the expectation. In the case when the global frequency vector is a low-degree polynomial, hybrid sketch performs similarly to fresh sketch, and both are better than shared sketch. As long as the global frequency vector is a slightly higher degree polynomial (e.g., with a degree higher than 3), then hybrid sketch is significantly better than both shared and fresh sketches.

$\mathbf{x}_t^{(m)} = \mathbf{e}_{x_t^{(m)}}$ where $(\mathbf{e}_t)_{t=1}^d$ refers to the canonical basis. Then the frequency vector can be denoted by $\mathbf{f} := (f_1, \ldots, f_d)^\top$. The aim is to estimate the frequency vector $\mathbf{f}$ in a manner that is compatible with SecSum.

**Baseline Method 1: Shared Sketch.** A multi-round FFE problem can be reduced to a single-round FFE problem with a large communication. Specifically, one can apply the CountSketch with the same randomness for every round; after collecting all the sketches from the $M$ round, one simply averages them. Due to the linearity of the sketching compress method, this is equivalent to a single round setting with $N = Mn$ clients. We refer to this method as *count sketch with shared hash design* (SharedSketch).

Thanks to the reduction idea, we can obtain the error and sketch size bounds for SharedSketch via applying Corollaries 2.2 and 2.3 to SharedSketch by replacing $n$ by $N = Mn$,

**Baseline Method 2: Fresh Sketch.** A multi-round FFE problem can also be broken down to $M$ independent single-round FFE problems. Specifically, one can apply *independent* CountSketch in each round, and decode $M$ local estimators for the $M$ local frequency vectors. As the CountSketch produces an unbiased estimator, one can show that the average of the $M$ local estimators is an unbiased estimator for the global frequency vector. We call this method *count sketch with fresh hash design* (FreshSketch). We provide the following bound for FreshSketch. The proof of which is motivated by Huang et al. [2021].

**Theorem 3.1** (Instance-specific bound for FreshSketch). *Let $(\hat{f}_j)_{j=1}^d$ be estimates produced by* FreshSketch. *Then for each $p \in (0, 1)$, $W \geq 1$ and $L \geq \log(1/p)$, it holds that: for each $j \in [d]$, with probability at least $1 - p$,*

$$|\hat{f}_j - f_j| < C \cdot \sqrt{\frac{\log(1/p)\log(M/p)}{L} \cdot \frac{1}{W} \cdot \sum_{i > W}(F_i^*)^2},$$

*where $C$ is an absolute constant, and $(F_i^*)_{i=1}^d$ are defined as in Theorem 3.2.*

**Hybrid Sketch.** Both SharedSketch and FreshSketch reduce a multi-round FFE problem into single-round FFE problem(s). In contrast, we show a more comprehensive sketching method, called *count sketch with hybrid hash design* (HybridSketch), that solves a multi-round FFE problem as a whole. HybridSketch is presented as Algorithm 2. Specifically, HybridSketch generates $M$ sketches that share a set of bucket hashes but use independent sets of sign hashes. Then in the $m$-th communication round, participating clients and the server communicate by the CountSketch algorithm based on the $m$-th sketch, so the server observes the summation of the sketched data through SecSum. After collecting $M$ summations of the sketched local data, the server first computes averages over different rounds for *variance reduction*, then computes the median over different repeats

---

**Algorithm 2** HYBRID SKETCH FOR FEDERATED FREQUENCY ESTIMATION

---

**Require:** The number of rounds $M$. $N = Mn$ clients with local data $x_t^{(m)} \in [d]$ for $m \in [M]$ and $t \in [n]$. Sketch length $L$ and width $W$.

1: The server prepares independent hash functions and broadcasts them to each client:

$$h_\ell : [d] \to [W], \ \sigma_\ell^{(m)} : [d] \to \{\pm 1\} \text{ for } \ell \in [L], \ m \in [M].$$

2: **for** Round $m = 1, \ldots, M$ in parallel **do**
3:     **for** Client $t = 1, \ldots, n$ in parallel **do**
4:         Client $(m, t)$ encodes the local data $x_t^{(m)}$ to $\mathtt{enc}^{(m)}\big(x_t^{(m)}\big) \in \mathbb{R}^{L \times W}$ where

$$\left(\mathtt{enc}^{(m)}\big(x_t^{(m)}\big)\right)_{\ell,k} = \mathbb{1}\left[h_\ell(x_t^{(m)}) = k\right] \cdot \sigma_\ell^{(m)}(x_t^{(m)}) \text{ for } \ell \in [L], \ k \in [W].$$

5:         Client $(m, t)$ sends $\mathtt{enc}^{(m)}\big(x_t^{(m)}\big)$ to $\mathtt{SecSum}$.
6:     **end for**
7:     $\mathtt{SecSum}$ receives $\big(\mathtt{enc}^{(m)}(x_t^{(m)})\big)_{t=1}^n$ and reveals the sum $\sum_{t=1}^n \mathtt{enc}^{(m)}\big(x_t^{(m)}\big)$ to the server.
8: **end for**
9: **for** Item $j = 1, \ldots, d$ in parallel **do**
10:     Server produces $M \times L$ estimators for $f_j$:

$$\mathtt{dec}(j; m, l) := \sigma_\ell^{(m)}(j) \cdot \left(\tfrac{1}{n} \sum_{t=1}^n \mathtt{enc}^{(m)}\big(x_t^{(m)}\big)\right)_{\ell, h_\ell(j)} \text{ for } m \in [M], \ell \in [L].$$

11:     Server computes the median over $\ell \in [L]$ of the averages over $m \in [M]$ of the estimators:

$$\mathtt{dec}(j) := \mathtt{median}\big\{\tfrac{1}{M} \sum_{m=1}^M \mathtt{dec}(j; m, l), \ \ell \in [L]\big\}.$$

12: **end for**
13: **return** $\big(\mathtt{dec}(j)\big)_{j=1}^d$ as estimate to $(f_j)_{j=1}^d$.

---

(or sketch rows) for *success probability amplification*. We provide the following problem-dependent bound for $\mathtt{HybridSketch}$.

**Theorem 3.2** (Instance-specific bound for $\mathtt{HybridSketch}$). *Let* $(\hat{f}_j)_{j=1}^d$ *be estimates produced by* $\mathtt{HybridSketch}$ *(see Algorithm 2). Define a* heterogeneity *vector* $(F_i)_{i=1}^d$ *by*

$$F_i := \frac{1}{M} \sqrt{\sum_{m=1}^M \big(f_i^{(m)}\big)^2}, \quad i = 1, \ldots, d.$$

*Clearly, it holds that* $F_i \le f_i$ *for every* $i \in [d]$. *Let* $(F_i^*)_{i \ge 1}$ *be* $(F_i)_{i \ge 1}$ *sorted in non-increasing order. Then for each* $p \in (0, 1)$, $W \ge 1$ *and* $L \ge \log(1/p)$, *it holds that: for each* $j \in [d]$, *with probability at least* $1 - p$,

$$|\hat{f}_j - f_j| < C \cdot \sqrt{\frac{\log(1/p)}{L} \cdot \frac{1}{W} \cdot \sum_{i > W}(F_i^*)^2},$$

*where* $C$ *is an absolute constant.*

We would like to point out that, although our $\mathtt{HybridSketch}$ algorithm is developed for multi-round frequency estimation problems, it can be adapted to multi-round vector recovery problems as well. Hence it could have broader applications in other federated learning scenarios.

**Hybrid Sketch vs. Fresh Sketch.** By comparing Theorem 3.2 with Theorem 3.1, we see that, with the same sketch size, the estimation error of $\mathtt{HybridSketch}$ is smaller than that of $\mathtt{FreshSketch}$ by a factor of $\sqrt{\log(M/p)}$. This provides theoretical insights that $\mathtt{HybridSketch}$ is superior to $\mathtt{FreshSketch}$ in terms of adapting to the instance hardness in multi-round FFE settings. This is also verified empirically by Figure 2.

**Hybrid Sketch vs. Shared Sketch.** We now compare the performance of `HybridSketch` and `SharedSketch` by comparing Theorem 3.2 and Proposition 2.1 (under a revision of replacing $n$ with $N = Mn$). Note that

$$F_i = \frac{1}{M} \sqrt{\sum_{m=1}^{M} \left( f_i^{(m)} \right)^2} \leq \frac{1}{M} \sum_{m=1}^{M} f_i^{(m)} = f_i.$$

So with the same sketch size, `HybridSketch` achieves an error that is no worse than csc in every case. Moreover, in the *homogeneous* case where all local frequency vectors are equivalent to the global frequency vector, i.e., $\mathbf{f}^{(m)} \equiv \mathbf{f}$ for all $m$, then it holds that $F_i = f_i / \sqrt{M}$. So in the homogeneous case, `HybridSketch` achieves an error that is smaller than that of csc by a factor of $1/\sqrt{M}$. In the general cases, the local frequency vectors are not perfectly homogeneous, then the improvement of `HybridSketch` over `SharedSketch` will depend on the *heterogeneity* of these local frequency vectors.

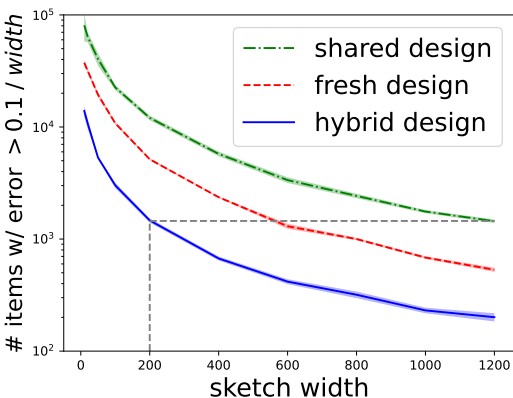

Figure 3: The number of items with error greater than 0.1/width for Shared, Hybrid, and Fresh Sketches with C4 dataset. HybridSketch with a width of 200 achieves roughly the same error as SharedSketch with a width of 1200 and Fresh sketch with a width of 600.

**Experiments.** We conduct three sets of experiments to verify our understandings about these sketches methods for multi-round FFE.

In the first sets of experiments, we simulate a multi-round FFE problem in homogeneous settings, where in every round the local frequency vectors are exactly the same. More specially, we set a domain size $d = 10^5$, a number of rounds $M = 10$ and test three different cases, where all the local frequency vectors are the same and (hence also the global frequency vector) are proportional to $(i^{-1.1})_{i=1}^{d}$, $(i^{-2})_{i=1}^{d}$ and $(i^{-5})_{i=1}^{d}$, respectively. In all the settings, we fix the sketch length to $L = 5$. In each experiment, we measure the expected $\ell_{\infty}$-error of each method with the averaging over $1,000$ independent repeats. The results are plotted in Figure 2. We can observe that: for low-degree polynomials, `HybridSketch` is nearly as good as `FreshSketch` and both are better than `SharedSketch`. But for slightly high degree polynomials (with a degree of 3), `HybridSketch` already outperforms both `FreshSketch` and `SharedSketch`. The numerical results are consistent with our theoretical analysis.

In the second sets of experiments, we simulate a multi-round FFE problem with the Gowalla dataset [Cho et al., 2011]. Similar to previously, we construct a domain of size $d = 175,000$, which corresponds to a grid over the US map. Then we sample $N = d = 175,000$ lists of the location information (that all belong to the domain created) to represent the data of $N$ clients, uniformly at random. We set the number of rounds to be $M = 10$. In each round, $n = N/M = 17,500$ clients participate. The results are presented in Figure 1(c). Here, the frequency and heterogeneity vectors have heavy tails, so `HybridSketch` and `FreshSketch` perform similarly and both are better than `SharedSketch`. This is consistent with our theoretical understanding.

In the third sets of experiments, we run simulations on the C4 [Bowman et al., 2020] dataset. Similar to the single round simulation, the domain size $d = 150,868$. We randomly sample $N = 150,000$ users from the dataset. The number of rounds $M = 10$, and in each round, $n = N/10 = 15,000$ clients participate. The results are provided in Figures 1(f) and 3. Here, the frequency and heterogeneity vectors have moderately light tails, and Figure 3 already suggests that `HybridSketch` produces an estimate that has a better shape than that produced by `FreshSketch` and `SharedSketch`, verifying the advantages of `HybridSketch`.

## 4 Differentially Private Sketches

While `SecSum` provides security guarantees, it does not provide differential privacy guarantees. In this part, we discuss a simple modifications to the sketching algorithms to make them provably differentially private (DP).

**Definition 2** (($\epsilon, \delta$)-DP [Dwork et al., 2006]). Let $\texttt{alg}(\cdot)$ be a randomized algorithm that takes a dataset $\mathcal{D}$ as its input. Let $\mathbb{P}$ be its probability measure. $\texttt{alg}(\cdot)$ is ($\epsilon, \delta$)-DP if: for every pair of neighboring datasets $\mathcal{D}$ and $\mathcal{D}'$, it holds that

$$\mathbb{P}\{\texttt{alg}(\mathcal{D}) \in \mathcal{E}\} < e^\epsilon \cdot \mathbb{P}\{\texttt{alg}(\mathcal{D}') \in \mathcal{E}\} + \delta.$$

In our case, a dataset corresponds to all participated clients (or their data), and two neighboring datasets should be regarded as two sets of clients (local data) that only differ in a single client (local data). The algorithm refers to all procedures before releasing the final frequency estimate, and all the intermediate computation is considered private and is not released.

We work with *central DP*, that is, server releases data in a differentially private way while clients do not release data. We focus on $\texttt{HybridSketch}$ as a representative algorithm. The DP mechanism can also be extended to the other sketching algorithms. Specifically, we use a DP mechanism that adds independent Gaussian noise to each entry of the sketching matrix, which is initially proposed for making $\texttt{CountSketch}$ differentially private by Pagh and Thorup [2022], Zhao et al. [2022].

We provide the following theorem characterizing the trade-off between privacy and accuracy.

**Theorem 4.1** (DP-hybrid sketch). *Consider a modified Algorithm 2, where we add to each entry of the sketching matrix an independent Gaussian noise, $\mathcal{N}(0, c_0 \cdot \sqrt{L \log(1/\delta)}/\epsilon)$, where $c_0 > 0$ is a known constant. Suppose that $L = \log(d/p)$ and $W \geq 2$. Then the final output of the modified Algorithm 2, denoted by $(\hat{f}_j)_{j=1}^d$, is ($\epsilon, \delta$)-DP for $\epsilon < 1$ and $\delta < 0.1$. Moreover, with probability at most $1 - p$, it holds that*

$$\max_j |\hat{f}_j - f_j| < C \cdot \left( \sqrt{\frac{\sum_{i>W}(F_i^*)^2}{W}} + \frac{\sqrt{\log(d/p)\log(1/\delta)}}{n\sqrt{M}\epsilon} \right),$$

*where $C > 0$ is an absolute constant and $(F_i^*)_{i=1}^d$ are as defined in Theorem 3.2.*

It is worth noting that if the number of clients per round ($n$) is fixed, then a larger number of rounds $M$ improves both the estimation error and the DP error in non-worst cases, e.g., when the local frequency vectors are nearly homogenous. However, if the total number of clients ($N = Mn$) is fixed, then a larger number of rounds $M$ improves the estimation error but makes the DP error worse.

When $M = 1$, Theorem 4.1 recovers the bounds for differentially private $\texttt{CountSketch}$ in Pagh and Thorup [2022], Zhao et al. [2022] and Theorem 5.1 in Chen et al. [2022]. Moreover, Chen et al. [2022] shows that in single-round FFE, for any algorithm that achieves an $\ell_\infty$-error smaller than $\tau := \mathcal{O}(\sqrt{\log(d)\log(1/\delta)}/(n\epsilon))$, in the worse case, each client must communicate $\Omega(n \cdot \min\{\sqrt{\log(d)/\log(1/\delta)}, \log(d)\})$ bits (see Their Corollary 5.1). In comparison, According to Theorem 4.1 and Corollary 2.3, the differentially private $\texttt{CountSketch}$ can achieve an $\ell_\infty$-error smaller than $\tau$ with length $L \approx \log(d)$ and width

$$W = C \cdot \min \left\{ \left( \#\{f_i : f_i \geq \tau\} + \frac{1}{\tau^2} \cdot \sum_{f_i < \tau} f_i^2 \right), n \right\} \leq C \cdot \min\{2/\tau, n\},$$

resulting in a per-client communication of $\mathcal{O}(WL\log(n))$ bits, which matches the minimax lower bound in Chen et al. [2022] ignoring a $\log(n)$ factor, but could be much smaller in non-worst cases where $(f_i)_{i=1}^d$ decays fast.

## 5 Concluding Remarks

We make several novel extensions to the count sketch method for federated frequency estimation with one or more communication rounds. In the single round setting, we show that count sketch can achieve better communication efficiency when the underlying problem is simpler. We provide a two-phase approach to automatically select a sketch size that adapts to the hardness of the problem. In the multiple rounds setting, we show a new sketching method that provably achieves better accuracy than simple adaptions of count sketch. Finally, we adapt the Gaussian mechanism to make the hybrid sketching method differentially private.

We remark that the improvement of the instance-dependent method relies on the assumptions that the underlying frequency has a lighter tail, which might be unverifiable a priori due to constraints, e.g., limited communication and privacy budget. Finally, this work focuses on an offline setting where the frequency is considered to be fixed. Extending our results to an online setting where the frequency is varying is an interesting future direction.

## Acknolwdgement

We thank the anonymous reviewers for their helpful comments. We thank Brendan McMahan for insightful discussions during the project. VB has been partially supported by National Science Foundation Awards 2244899 and 2333887 and the ONR award N000142312737.

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

# A  Missing Proofs for Section 2

## A.1  Proof of Proposition 2.1

*Proof of Proposition 2.1.* We refer the reader to Theorem 4.1 in Minton and Price [2014]. □

## A.2  Proof of Corollary 2.2

*Proof of Corollary 2.2.* From Proposition 1 we know that

$$\text{for every } j \in [d], \quad \mathbb{P}\left\{|\text{dec}(j) - f_j| > C \cdot \sqrt{\frac{\log(1/\delta)}{L} \cdot \frac{1}{W} \cdot \sum_{i>W}(f_i^*)^2}\right\} < \delta.$$

By union bound we have

$$\mathbb{P}\left\{\text{there exists } j \in [d], |\text{dec}(j) - f_j| > C \cdot \sqrt{\frac{\log(1/\delta)}{L} \cdot \frac{1}{W} \cdot \sum_{i>W}(f_i^*)^2}\right\} < d\delta.$$

Replacing $\delta$ with $\delta/d$, setting $L = \log(d/\delta)$, and using the definition of $\ell_\infty$-norm, we obtain

$$\mathbb{P}\left\{\|\text{dec}(\cdot) - \mathbf{f}\|_\infty > C \cdot \sqrt{\frac{1}{W} \cdot \sum_{i>W}(f_i^*)^2}\right\} < \delta.$$

We next show that:

$$\sqrt{\frac{1}{W} \cdot \sum_{i>W}(f_i^*)^2} \le \frac{1}{W}.$$

To this end, we first show that $f_W^* \le \frac{1}{W}$. If not, we must have for $i = 1, \ldots, W$, $f_i^* \ge f_W^* > \frac{1}{W}$, as $(f_i^*)_{i=1}^d$ is sorted in non-increasing order. Then $\sum_i^d f_i^* \ge \sum_{i=1}^W f_i^* > 1$, which contradicts to the fact that $(f_i^*)_{i=1}^d$ is a frequency vector. We have shown that $f_W^* \le \frac{1}{W}$, and this further implies that for any $i \ge W$, $f_i^* \le f_W^* \le \frac{1}{W}$. Then we can obtain

$$\sqrt{\frac{1}{W} \cdot \sum_{i>W}(f_i^*)^2} \le \sqrt{\frac{1}{W^2} \cdot \sum_{i>W} f_i^*} \le \frac{1}{W},$$

since $(f_i^*)_{i=1}^d$ is a frequency vector. We have completed all the proof. □

## A.3  Proof of Corollary 2.3

*Proof of Corollary 2.3.* Define

$$E(W) := \sqrt{\frac{1}{W} \sum_{i>W}(f_i^*)^2}.$$

We will show the following:

1. If $W \ge \#\{f_i \ge \tau\} + \frac{1}{\tau^2}\sum_{f_i<\tau} f_i^2$, then $E(W) \le \tau$.

2. Moreover, if $E(W) \le \tau$, then $W \ge \frac{1}{2}\left(\#\{f_i \ge \tau\} + \frac{1}{\tau^2}\sum_{f_i<\tau} f_i^2\right)$.

Then Corollary 2.3 follows by combining Corollary 2.2 with the above claims.

We first show the first part. First note that $W \ge \#\{f_i \ge \tau\}$ and that $(f_i^*)_{i=1}^d$ is sorted in non-increasing order, so for all $i \ge W$ it holds that $f_i^* < \tau$. Therefore,

$$E(W) := \sqrt{\frac{1}{W} \sum_{i>W}(f_i^*)^2} \le \sqrt{\frac{1}{W} \sum_{f_i<\tau} f_i^2}.$$

Moreover, note that $W \ge \frac{1}{\tau^2}\sum_{f_i<\tau} f_i^2$, so we further have $E(W) \le \tau$.

To show that second part, we first note that, by definition, $E(W) \le \tau$ is equivalent to

$$2W \ge W + \frac{1}{\tau^2} \sum_{i>W} (f_i^*)^2.$$

Consider the following function

$$F(k) := k + \frac{1}{\tau^2} \sum_{i>k} (f_i^*)^2, \quad k \ge 1,$$

one can directly verify that $F(k)$ is minimized at $k^* := \#\{i : f_i \ge \tau\}$; moreover,

$$F(k^*) = k^* + \frac{1}{\tau^2} \sum_{i>k^*} (f_i^*)^2 = \#\{f_i \ge \tau\} + \frac{1}{\tau^2} \sum_{f_i < \tau} f_i^2.$$

Therefore, we have

$$2W \ge F(W) \ge F(k^*) = \#\{f_i \ge \tau\} + \frac{1}{\tau^2} \sum_{f_i < \tau} f_i^2.$$

This completes our proof. $\qquad\square$

# B    Missing Proofs for Section 3

## B.1    Proof of Theorem 3.1

*Proof of Theorem 3.1.*  The proof is motivated by Huang et al. [2021].

Define the following events

$$E_j^{(m)} := \left\{ |\hat{f}_j^{(m)} - f_j^{(m)}| \le C \cdot \sqrt{\frac{\log(1/p)}{L} \cdot \frac{1}{W} \cdot \sum_{i>W} (f_i^{(m)})^2} \right\}, \ m \in [M], j \in [d].$$

Then by Proposition 2.1 we have

$$\mathbb{P}\{E_j^{(m)}\} \ge 1 - p.$$

Then by union bound, we have

$$\mathbb{P}\left\{ \bigcap_{m=1}^{M} E_j^{(m)} \right\} \ge 1 - Mp.$$

Conditional on the event of $\bigcap_{m=1}^{M} E_j^{(m)}$, we know that every random variable $\hat{f}_j^{(m)} - f_j^{(m)}$ is bounded within

$$\left( -F^{(m)}, \ F^{(m)} \right),$$

where

$$F^{(m)} := C \cdot \sqrt{\frac{\log(1/p)}{L} \cdot \frac{1}{W} \cdot \sum_{i>W} \left(f_i^{(m)}\right)^2}.$$

So by Hoeffding inequality, we have

$$\mathbb{P}\left\{ \left| \frac{1}{M} \sum_{m=1}^{M} \hat{f}_j^{(m)} - \frac{1}{M} \sum_{m=1}^{M} f_j^{(m)} \right| \le \sqrt{\frac{\log(2/p_1)}{2M^2} \sum_{m=1}^{M} \left(F^{(m)}\right)^2} \ \middle| \ \bigcap_{m=1}^{M} E_j^{(m)} \right\} \ge 1 - p_1$$

Then we have

$$\mathbb{P}\left\{ \left| \frac{1}{M} \sum_{m=1}^{M} \hat{f}_j^{(m)} - \frac{1}{M} \sum_{m=1}^{M} f_j^{(m)} \right| \le \sqrt{\frac{\log(2/p_1)}{2M^2} \sum_{m=1}^{M} \left(F^{(m)}\right)^2} \right\} \ge 1 - p_1 - Mp.$$

Note that

$$\frac{\log(2/p_1)}{2M^2} \sum_{m=1}^{M} \left(F^{(m)}\right)^2 = \frac{\log(2/p_1)}{2M^2} \sum_{m=1}^{M} C^2 \cdot \frac{\log(1/p)}{L} \cdot \frac{1}{W} \cdot \sum_{i>W} \left(f_i^{(m)}\right)^2$$

$$= C^2 \cdot \frac{\log(2/p_1)\log(1/p)}{2L} \cdot \frac{1}{W} \cdot \sum_{i>W} \left(F_i\right)^2.$$

So we have

$$\mathbb{P}\left\{\left|\frac{1}{M} \sum_{m=1}^{M} \hat{f}_j^{(m)} - \frac{1}{M} \sum_{m=1}^{M} f_j^{(m)}\right| \le \sqrt{C^2 \cdot \frac{\log(2/p_1)\log(1/p)}{2L} \cdot \frac{1}{W} \cdot \sum_{i>W} \left(F_i\right)^2}\right\}$$

$$\ge 1 - p_1 - Mp.$$

Note replace $p_1 = p'/2$ and $p = p'/(2M)$, we have that

$$\mathbb{P}\left\{\left|\frac{1}{M} \sum_{m=1}^{M} \hat{f}_j^{(m)} - \frac{1}{M} \sum_{m=1}^{M} f_j^{(m)}\right| \le \sqrt{C' \cdot \frac{\log(1/p')\log(M/p')}{L} \cdot \frac{1}{W} \cdot \sum_{i>W} \left(F_i\right)^2}\right\} \ge 1 - p'.$$

$\square$

## B.2  Proof of Theorem 3.2

*Proof of Theorem 3.2.*  Let us consider the hybrid sketch approach in Algorithm 2. Recall that within a round, clients use the same set of hash functions to construct their sketching matrices. Across different rounds, clients use the same set of location hashes but a fresh set of sign hashes. Denote the hash functions by:

$$h_\ell : [d] \to [w], \quad \ell = 1, \dots, L;$$
$$\sigma_\ell^{(m)} : [d] \to \{+1, -1\}, \quad \ell = 1, \dots, L; \; m = 1 \dots, M.$$

Recall the local frequency in each round is defined by

$$\mathbf{f}^{(m)} := \frac{1}{n} \sum_{t=1}^{n} \mathbf{x}^{(m,t)}, \quad m = 1, \dots, M.$$

And the global frequency vector is defined by

$$\mathbf{f} := \frac{1}{M} \sum_{m=1}^{M} \mathbf{f}^{(m)}.$$

Then according to the communication protocol, the server receives $M$ sketching matrices (each corresponds to a summation of clients' sketches within the same round). From the $m$-th sketch, we can extract $L$ estimators for each index $j \in [d]$, i.e.,

$$\tilde{\mathbf{f}}_j^{(m,\ell)} := \sum_{i=1}^{d} \mathbb{1}\left[h_\ell(i) = h_\ell(j)\right] \cdot \sigma_\ell^{(m)}(j) \cdot \sigma_\ell^{(m)}(i) \cdot \mathbf{f}_i^{(m)}, \quad j \in [d], \; m \in [M], \; \ell \in [L]$$

$$= \mathbf{f}_j^{(m)} + \sum_{i \ne j} \mathbb{1}\left[h_\ell(i) = h_\ell(j)\right] \cdot \sigma_\ell^{(m)}(j) \cdot \sigma_\ell^{(m)}(i) \cdot \mathbf{f}_i^{(m)}.$$

For each index, we will first average the estimators from different rounds to reduce the variance, then take the median over different rows to amplify the success probability. In particular, denote the round-wise averaging by

$$\tilde{\mathbf{f}}_j^{(\ell)} := \frac{1}{M} \sum_{m=1}^{M} \tilde{\mathbf{f}}_j^{(m,\ell)}, \quad j \in [d], \; \ell \in [L]$$

$$= \frac{1}{M} \sum_{m=1}^{M} \mathbf{f}_j^{(m)} + \frac{1}{M} \sum_{m=1}^{M} \sum_{i \ne j} \mathbb{1}\left[h_\ell(i) = h_\ell(j)\right] \cdot \sigma_\ell^{(m)}(j) \cdot \sigma_\ell^{(m)}(i) \cdot \mathbf{f}_i^{(m)}$$

$$= \underbrace{\mathbf{f}_j}_{\texttt{signal}} + \underbrace{\frac{1}{M} \sum_{i \neq j} \mathbb{1}\left[h_\ell(i) = h_\ell(j)\right] \cdot \sum_{m=1}^{M} \sigma_\ell^{(m)}(j) \cdot \sigma_\ell^{(m)}(i) \cdot \mathbf{f}_i^{(m)}}_{\texttt{noise}}$$

$$= \underbrace{\mathbf{f}_j}_{\texttt{signal}} + \underbrace{\frac{1}{M} \sum_{i \neq j, i \in \mathbb{W}} \mathbb{1}\left[h_\ell(i) = h_\ell(j)\right] \cdot \sum_{m=1}^{M} \sigma_\ell^{(m)}(j) \cdot \sigma_\ell^{(m)}(i) \cdot \mathbf{f}_i^{(m)}}_{\texttt{headNoise}}$$

$$+ \underbrace{\frac{1}{M} \sum_{i \neq j, i \notin \mathbb{W}} \mathbb{1}\left[h_\ell(i) = h_\ell(j)\right] \cdot \sum_{m=1}^{M} \sigma_\ell^{(m)}(j) \cdot \sigma_\ell^{(m)}(i) \cdot \mathbf{f}_i^{(m)}}_{\texttt{tailNoise}}. \tag{5}$$

Then we take the median over these estimators to obtain

$$\tilde{\mathbf{f}}_j := \texttt{median}\{\tilde{\mathbf{f}}_j^{(\ell)}, \ \ell \in [L]\}, \quad j \in [d].$$

**Head Noise.** The only randomness comes from the algorithm. Note that the head noise contains at most $|\mathbb{W}| \leq 0.1W$ independent terms, and each is zero with probability $1 - 1/W$. Thus the head noise is zero with probability at least $(1 - 1/W)^{|\mathbb{W}|} \geq (1 - 1/W)^{0.1W} \geq 0.9$ provided that $W > 10$.

**Tail Noise.** Now consider the second noise term in (5). Fixing $\ell$ and $j$. Define

$$\xi_i^{(m)} := \sigma_\ell^{(m)}(j) \cdot \sigma_\ell^{(m)}(i) \cdot \mathbf{f}_i^{(m)}$$

$$\xi_i := \sum_{m=1}^{M} \xi_i^{(m)} = \sum_{m=1}^{M} \sigma_\ell^{(m)}(j) \cdot \sigma_\ell^{(m)}(i) \cdot \mathbf{f}_i^{(m)}$$

$$\eta_i := \mathbb{1}\left[h(i) = h(j)\right]$$

$$\texttt{tailNoise} := \frac{1}{M} \sum_{i \neq j, i \notin \mathbb{W}} \eta_i \cdot \xi_i.$$

First notice that $\left(\xi_i^{(m)}\right)_{m=1}^{M}$ are independent random variables and

$$\mathbb{E}[\xi_i^{(m)}] = 0, \quad \mathrm{Var}[\xi_i^{(m)}] = \left(\mathbf{f}_i^{(m)}\right)^2.$$

These imply that

$$\mathbb{E}[\xi_i] = 0, \quad \mathrm{Var}[\xi_i] = \sum_{m=1}^{M} \left(\mathbf{f}_i^{(m)}\right)^2.$$

Moreover, notice that $(\eta_i, \xi_i)_{i \neq j}$ are independent random variables, and

$$\mathbb{E}[\eta_i^2] = \frac{1}{W},$$

we then have

$$\mathbb{E}[\eta_i \xi_i] = 0;$$

$$\mathrm{Var}[\eta_i \xi_i] = \mathbb{E}[\eta_i^2] \cdot \mathrm{Var}[\xi_i] + \mathrm{Var}[\eta_i] \cdot \left(\mathbb{E}[\xi_i]\right)^2$$

$$= \frac{1}{W} \cdot \sum_{m=1}^{M} \left(\mathbf{f}_i^{(m)}\right)^2.$$

Therefore we conclude that

$$\mathbb{E}[\texttt{tailNoise}] = \frac{1}{M} \sum_{i \neq j, i \notin \mathbb{W}} \mathbb{E}[\eta_i \xi_i] = 0;$$

$$\mathrm{Var}[\texttt{tailNoise}] = \frac{1}{M^2} \sum_{i \neq j, i \notin \mathbb{W}} \mathrm{Var}[\eta_i \xi_i]$$

$$= \frac{1}{M^2 W} \cdot \sum_{i \neq j, i \notin \mathbb{W}} \sum_{m=1}^{M} \left(\mathbf{f}_i^{(m)}\right)^2$$

$$\leq \frac{1}{M^2 W} \cdot \sum_{i \notin \mathbb{W}} \sum_{m=1}^{M} \left(\mathbf{f}_i^{(m)}\right)^2.$$

Then by Chebyshev we see that: for fixed $j \in [d]$ and $\ell \in [L]$ it holds that

$$\mathbb{P}\left\{|\mathtt{tailNoise}| \geq \sqrt{\frac{10}{M^2 W} \cdot \sum_{i \notin \mathbb{W}} \sum_{m=1}^{M} \left(\mathbf{f}_i^{(m)}\right)^2}\right\} < 0.1.$$

By a union bound we see that: for fixed $j \in [d]$ and $\ell \in [L]$ it holds that

$$\mathbb{P}\left\{|\tilde{\mathbf{f}}_j^{(\ell)} - \mathbf{f}_j| < \sqrt{\frac{10}{M^2 W} \cdot \sum_{i \notin \mathbb{W}} \sum_{m=1}^{M} \left(\mathbf{f}_i^{(m)}\right)^2}\right\} > 0.8 > 0.5.$$

**Probability Amplification.** Fixing $j$. Recall that $(\tilde{\mathbf{f}}_j^{(\ell)})_{\ell=1}^{L}$ are i.i.d. random variables and that $\tilde{\mathbf{f}}_j := \mathtt{median}\{\tilde{\mathbf{f}}_j^{(\ell)} : \ell \in [L]\}$. By Chernoff over $\ell$ and union bound over $j$ we see that:

$$\mathbb{P}\left\{\text{for each } j \in [d], \quad |\tilde{\mathbf{f}}_j - \mathbf{f}_j| \geq \sqrt{\frac{10}{M^2 W} \cdot \sum_{i \notin \mathbb{W}} \sum_{m=1}^{M} \left(\mathbf{f}_i^{(m)}\right)^2}\right\} < 2d \cdot \exp(\Omega(L)).$$

By choosing $L = \Theta(\log(2d/\delta))$ we obtain that, with probability at least $1 - \delta$,

$$\text{for each } j \in [d], \quad |\tilde{\mathbf{f}}_j - \mathbf{f}_j| \lesssim \sqrt{\frac{10}{M^2 W} \cdot \sum_{i \notin \mathbb{W}} \sum_{m=1}^{M} \left(\mathbf{f}_i^{(m)}\right)^2}.$$

$\square$

## C  Missing Proofs for Section 4

### C.1  Proof of Theorem 4.1

*Proof of Theorem 4.1.* We follow the method of Pagh and Thorup [2022], Zhao et al. [2022] to add DP noise to all $M$ sketches. Suppose $\mathcal{F} = \left(\mathbf{f}^{(m)}\right)_{m=1}^{M}$ and $\mathring{\mathcal{F}} = \left(\mathring{\mathbf{f}}^{(m)}\right)_{m=1}^{M}$ are the sets of local frequencies for two neighboring datasets respectively, then

$$\|\mathcal{F} - \mathring{\mathcal{F}}\|_2 \leq \frac{1}{n}.$$

Denote the sketches to be released by $\mathcal{S} \circ \mathcal{F} := \left(\mathbf{S}^{(m)} \circ \mathbf{f}^{(m)}\right)_{m=1}^{M}$. One can then calculate the $\ell_2$-sensitivity:

$$\|\mathcal{S} \circ \mathcal{F} - \mathcal{S} \circ \mathring{\mathcal{F}}\|_2 \leq \frac{\sqrt{L}}{n},$$

where $L \approx \log(d/\delta)$ is the sketch length. Therefore the sketching will be $(\epsilon, \delta)$-DP by adding Gaussian noise $\mathcal{N}(0, \sigma^2)$ to each bucket of each sketch, where

$$\sigma \approx \frac{\sqrt{L \log(1/\delta)}}{n\epsilon}.$$

The final released frequency estimator is obtained by post-processing the sketch, so it is also $(\epsilon, \delta)$-DP.

We then calculate the error for the noisy sketch matrix. For each row estimator, we have that with probability at least $2/3$:

$$\tilde{\mathbf{f}}_j^{(\ell)} - \mathbf{f}_j^{\ell} = \mathtt{tailNoise} + \frac{1}{M} \sum_{m=1}^{M} \mathtt{rad}_m \cdot \mathcal{N}(0, \sigma^2)$$

$$= \texttt{tailNoise} + \mathcal{N}(0, \sigma^2/M)$$

$$\lesssim \sqrt{\frac{1}{M^2 w} \cdot \sum_{i \notin \mathbb{W}} \sum_{m=1}^{M} \left(\mathbf{f}_i^{(m)}\right)^2} + \frac{\sqrt{L \log(1/\delta)}}{\sqrt{M} n \epsilon}.$$

By taking median over $L \asymp \log(d/\delta)$ repeats, we see that with probability at least $1 - \delta$, it holds that

$$\text{for each } j \in [d], \quad |\hat{\mathbf{f}}_j - \mathbf{f}_j| \lesssim \sqrt{\frac{1}{M^2 w} \cdot \sum_{i \notin \mathbb{W}} \sum_{m=1}^{M} \left(\mathbf{f}_i^{(m)}\right)^2} + + \frac{\sqrt{\log(d/\delta) \cdot \log(1/\delta)}}{\sqrt{M} n \epsilon}.$$

$\square$

