# Private Federated Frequency Estimation: Adapting to the Hardness of the Instance

## Abstract

In *federated frequency estimation* (FFE), multiple clients work together to estimate the frequency of their local data by communicating with a server, while respecting the security constraints of Secure Summation (`SecSum`) where the server can only access the sum of client-held vectors. For FFE with a single communication round, it is known that *count sketch* is nearly information-theoretically optimal [8]. However, when multiple communication rounds are allowed, we propose a new sketching algorithm that is *provably* more accurate than a naive adaptation of count sketch. Furthermore, we show that both our sketch algorithm and count sketch can achieve better accuracy when the problem instance is simpler. Therefore, we propose a two-phase approach to enable the use of a smaller sketch size for simpler problems. Finally, we provide mechanisms to make our proposed algorithm differentially private. We verify the superior performance of our methods through experiments conducted on several largescale datasets.

## 1 Introduction

In many distributed learning applications, a server seeks to compute population information about data that is distributed across multiple clients (users). For example, consider a distributed frequency estimation problem where there are $n$ clients, each holding a local data from a domain of size $d$, and a server that aims to estimate the frequency of the items from the $n$ clients with a minimum communication cost. This task can be done efficiently by letting each client *binary encode* their data and send the encoding to the server, at a local communication bandwidth cost of $\log(d)$ bits. Provided with the binary encoding, the server can faithfully decode *each* local data and compute the global frequency vector (i.e., the normalized histogram vector).

However, the local data could be sensitive or private, and the clients may wish to keep it hidden from the server. The above binary encoding communication method, unfortunately, allows the server to observe each individual local data, and therefore may not satisfy the users' privacy concerns. *Federated Analytics* (FA) [16, 18] addresses this issue by developing new methods that enable the server to learn population information about the clients while preventing the server from prying on any individual local data. In particular, a cryptographic multi-party computation protocol, *Secure Summation* (`SecSum`) [1], has become a widely adopted solution to provide data minimization guarantees for FA [3]. Specifically, `SecSum` sets up a communication protocol between clients and the server, which injects carefully designed additive noise to each data that cancels out when *all of the local data is summed together*, but blurs out (information theoretically) each individual local data otherwise. Under `SecSum`, the server is able to faithfully obtain the correct summation of the data from all clients but is unable to read a single local data. *Federated frequency estimation* (FFE) problems refer to the distributed frequency estimation problems under the constraint of `SecSum`. Clearly, the binary encoding method is not compatible with `SecSum`, because when the binary encoding is passed

to the server through `SecSum`, the server only gets the summation of the binary encodings of the users' data, which does not provide sufficient information for computing the global frequency vector.

A naive approach to FFE is by employing *one-hot encoding*: each client encodes its local data into a $d$-dimensional one-hot vector that represents the local frequency vector and sends it to the server through `SecSum`. Then the server observes the summation of the local frequency vectors using `SecSum` and scales it by the number of clients to obtain the true frequency vector. However, the one-hot encoding approach costs $\Theta(d \log(n))$ bits of communication bandwidth. This is because `SecSum` adds noise from a field of size $\Theta(n)$ to each component of the $d$-dimensional local frequency vector [1]. With a linear dependence on domain size $d$, the one-hot encoding approach is inefficient for large domain problems, especially when the domain size exceeds the number of clients ($d > n$). In what follows, we will focus on this regime and assume that $

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

5: **end for**
6: $\texttt{SecSum}$ receives $\big(\texttt{enc}(x_t)\big)_{t=1}^{n}$ and reveals the summation $\sum_{t=1}^{n} \texttt{enc}(x_t)$ to the server.
7: **for** Item $j = 1, \dots, d$ in parallel **do**
8:     Server produces $L$ estimators for $f_j$:

$$\texttt{dec}(j; \ell) := \sigma_\ell(j) \cdot \big(\tfrac{1}{n} \sum_{t=1}^{n} \texttt{enc}(x_t)\big)_{\ell, h_\ell(j)} \text{ for } \ell \in [L].$$

9:     Server computes the median of the $L$ estimators:

$$\texttt{dec}(j) := \texttt{median}\{\texttt{dec}(j; \ell) : \ell \in [L]\}.$$

10: **end for**
11: **return** $(\texttt{dec}(j))_{j=1}^{d}$ as estimate to $(f_j)_{j=1}^{d}$.

---

## B   Additional Experiments

**Sentiment-140.**   We also run additional simulations on a Twitter dataset Sentiment-140 [11]. The dataset contains $d = 739,972$ unique words from $N = 659,497$ users. We randomly sample one word from each user to construct our experiment dataset. The number of rounds $M = 10$, and in each round, $n = N/10 = 65,949$ clients participate. Results are provided in Figure 4.

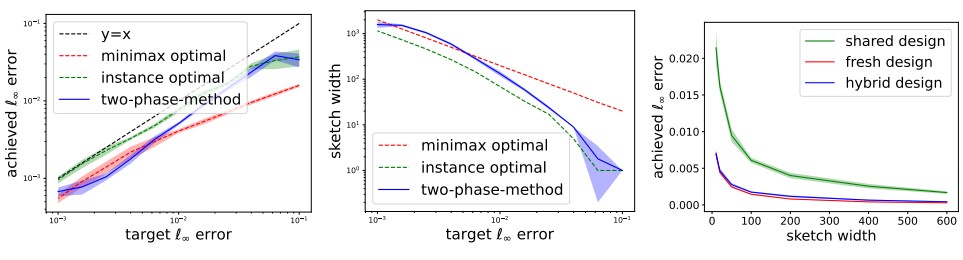

(a) Setiment 140, single round    (b) Setiment 140, single round    (c) Setiment 140, multi-round

Figure 4: Single-round and multi-round FFE simulations on the Sentiment-140 dataset.

**Additional Plots for Single-Round FFE.**   Figure 5 provides some additional results in our single-round FFE simulations.

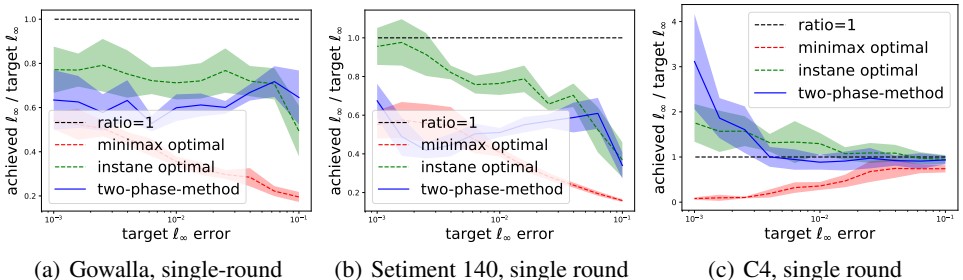

(a) Gowalla, single-round    (b) Setiment 140, single round    (c) C4, single round

Figure 5: Single-round federated frequency estimation experiments.

## C  Missing Proofs for Section 2

### C.1  Proof of Proposition 2.1

*Proof of Proposition 2.1.*  We refer the reader to Theorem 4.1 in Minton and Price [13].  □

### C.2  Proof of Corollary 2.2

*Proof of Corollary 2.2.*  From Proposition 1 we know that

$$\text{for every } j \in [d], \quad \mathbb{P}\left\{|\mathtt{dec}(j) - f_j| > C \cdot \sqrt{\frac{\log(1/\delta)}{L} \cdot \frac{1}{W} \cdot \sum_{i>W}(f_i^*)^2}\right\} < \delta.$$

By union bound we have

$$\mathbb{P}\left\{\text{there exists } j \in [d], \ |\mathtt{dec}(j) - f_j| > C \cdot \sqrt{\frac{\log(1/\delta)}{L} \cdot \frac{1}{W} \cdot \sum_{i>W}(f_i^*)^2}\right\} < d\delta.$$

Replacing $\delta$ with $\delta/d$, setting $L = \log(d/\delta)$, and using the definition of $\ell_\infty$-norm, we obtain

$$\mathbb{P}\left\{\|\mathtt{dec}(\cdot) - \mathbf{f}\|_\infty > C \cdot \sqrt{\frac{1}{W} \cdot \sum_{i>W}(f_i^*)^2}\right\} < \delta.$$

We next show that:

$$\sqrt{\frac{1}{W} \cdot \sum_{i>W}(f_i^*)^2} \leq \frac{1}{W}.$$

To this end, we first show that $f_W^* \leq \frac{1}{W}$. If not, we must have for $i = 1, \ldots, W$, $f_i^* \geq f_W^* > \frac{1}{W}$, as $(f_i^*)_{i=1}^d$ is sorted in non-increasing order. Then $\sum_i^d f_i^* \geq \sum_{i=1}^W f_i^* > 1$, which contradicts to the fact that $(f_i^*)_{i=1}^d$ is a frequency vector. We have shown that $f_W^* \leq \frac{1}{W}$, and this further implies that for any $i \geq W$, $f_i^* \leq f_W^* \leq \frac{1}{W}$. Then we can obtain

$$\sqrt{\frac{1}{W} \cdot \sum_{i>W}(f_i^*)^2} \leq \sqrt{\frac{1}{W^2} \cdot \sum_{i>W} f_i^*} \leq \frac{1}{W},$$

since $(f_i^*)_{i=1}^d$ is a frequency vector. We have completed all the proof.  □

### C.3  Proof of Corollary 2.3

*Proof of Corollary 2.3.*  Define

$$E(W) := \sqrt{\frac{1}{W}\sum_{i>W}(f_i^*)^2}.$$

We will show the following:

1. If $W \geq \#\{f_i \geq \tau\} + \frac{1}{\tau^2}\sum_{f_i<\tau} f_i^2$, then $E(W) \leq \tau$.