# OpenReview forum: "Private Federated Frequency Estimation: Adapting to the Hardness of the Instance"
_NeurIPS.cc/2023/Conference — NeurIPS 2023 poster_

### Official Review · Reviewer_q1hW · 2023-06-17

**Soundness:** 3 good
**Presentation:** 3 good
**Contribution:** 3 good
**Rating:** 7
**Confidence:** 5

**Summary:**

In the nascent area of 'federated analytics', the canonical problem is frequency estimation: each client holds a label (or a collection of labels), and the aim is to recover the frequency distribution of these labels, or at least information on the most frequent labels (heavy hitters).  Prior work has demonstrated that this problem can be tackled by making use of 'sketch' data structures designed over the last few years.  The question is now to understand how to optimize the accuracy-communication tradeoff, while meeting various security and privacy guarantees.  This paper builds on prior work in this direction, by arguing that better bounds can be achieved due to the skewed nature of the input data: essentially, the worst case for sketches is when the data is uniform, and better accuracy is seen when most mass is in the head of the label frequency distribution.  Results for this case follow in part due to prior analysis of such data structures, adapted to the parameters of the federated setting. The second results apply to the case when data is collected in multiple rounds, by comparing the effect of using the same randomly chosen sketch parameters versus varying some.  The results show that this achieves improved bounds in the case that the frequency distribution changes between rounds.  Experiments based on simulated data allocations quantify this further.

**Strengths:**

This is a foundational problem for the area of federated computing, and the paper shows some improved results which could be of use to practitioners.

The algorithms proposed are clear and suitable to be implemented.  It is straightforward to achieve differential privacy by noise addition to the sketch (under secure aggregation), with bounded impact on accuracy.

**Weaknesses:**

The novel contribution is not extremely high.  The first set of results are shown by plugging parameters into theorems from prior work, and some manipulation of probabilities. The second set of results are more involved, but can still be viewed as adaptations of prior proofs.  From a technical perspective, there is not much excitement.  The potential practical implications could elevate this work, but this would require more effort to demonstrate that the real world scenarios where this is needed map on to the assumptions in this work.

The results separate best from the single sketch approach when the data is 'heterogenous'.  The paper could do more to define the exact model of heterogeneity assumed, and to test these on real data.  The main opportunity for hybrid sketch to shine (P8) is when the frequency vectors are heterogenous -- that is, when the frequency distribution is different during each round of data collection.  (Note that most commonly, in the federated setting, heterogenous is used to refer to the case when the allocating of data to each client can vary a lot.  However, the definition of the frequency estimation problem is oblivious to the allocation of labels to clients: it just asks for the global frequency distribution). It would be helpful to know what kind of scenarios expect to have different frequency distributions per round, and what are the implications of this for the frequency estimation problem -- when the data is constantly shifting, what should the ground truth frequency distribution be?

**Questions:**

Are there any good data sets or references that can help to argue that the heterogeneity vectors F are substantially smaller (in Euclidean norm) than the frequency vectors?

**Limitations:**

No limitations in the send of negative societal impact
Could say more about technical limitations of the work, or future work -- the conclusion just summarizes the results.

---

> ### Author Rebuttal · Authors · 2023-08-08
>
> Thank you for your comments and suggestions. In the following, we will address your questions.
>
> ---
> **Q1**. “The novel contribution is not extremely high. The first set of results are shown by plugging parameters into theorems from prior work, and some manipulation of probabilities. The second set of results are more involved, but can still be viewed as adaptations of prior proofs. From a technical perspective, there is not much excitement. The potential practical implications could elevate this work, but this would require more effort to demonstrate that the real world scenarios where this is needed map on to the assumptions in this work.”
>
> **A1**. We use some standard techniques to derive our bounds. However, we’d like to emphasize that both the considered multi-round FFE problem settings and results are new. It also requires non-trivial efforts to establish new effective algorithms such as Hybrid Sketch, to allow the sketching method to adapt to the hardness of the instance, and to make the algorithm differentially private.
>
> ---
> **Q2**. “The results separate best from the single sketch approach when the data is 'heterogenous'. The paper could do more to define the exact model of heterogeneity assumed, and to test these on real data. The main opportunity for hybrid sketch to shine (P8) is when the frequency vectors are heterogenous -- that is, when the frequency distribution is different during each round of data collection. (Note that most commonly, in the federated setting, heterogenous is used to refer to the case when the allocating of data to each client can vary a lot. However, the definition of the frequency estimation problem is oblivious to the allocation of labels to clients: it just asks for the global frequency distribution). It would be helpful to know what kind of scenarios expect to have different frequency distributions per round, and what are the implications of this for the frequency estimation problem -- when the data is constantly shifting, what should the ground truth frequency distribution be?”
>
> **A2**. We'd like to make two clarifications regarding our results.
>
> Firstly, Hybrid Sketch separates best from Shared Sketch when the multi-round dataset is *homogenous* rather than heterogenous (see the discussions in lines 247-264). Therefore, the main opportunity for Hybrid Sketch to shine is when local frequency vectors are closer to homogenous rather than heterogeneous.
>
> Secondly, our definition of (single-round) frequency estimation problems only asks for a frequency vector. However, our definition of multi-round frequency estimation problems asks for local frequency vectors in each round instead of only a global frequency vector (this can be seen from Theorem 3.2). So the latter problem is *not* oblivious to data allocation.
>
> We hope we have clarified the questions. Please let us be aware if there are any further concerns.
>
> The online setting where data is constantly shifting is very interesting. However, our main focus in this work is an ‘’offline’’ setting where the number of rounds $M$ is fixed (and known). We will comment on the online setting as a future work.
>
> ---
> **Q3**. “Are there any good data sets or references that can help to argue that the heterogeneity vectors F are substantially smaller (in Euclidean norm) than the frequency vectors?”
>
> **A3**. In an ideal case where the local frequency vectors are perfectly homogenous, we can show that (see lines 247-264)
> $$ F_i = \frac{1}{\sqrt{M}} f_i,$$
> where $M$ is the number of rounds. So the heterogeneity vector is only $1/\sqrt{M}$ of the global frequency vector, pointwise, hence also in Euclidean norm. In practice, the multi-round datasets are often *approximately* homogeneous, so we expect the heterogeneity vector to be substantially smaller than the global frequency vector.

---

> > ### Comment · Reviewer_q1hW · 2023-08-11
> >
> > Thank you for the careful response, and the clarification around the interpretation of heterogeneity in this work.  The empirical results show that there is a clearer separation when there is a lot of skew in the frequency distribution (Fig 2b and 2c) -- it would be good to comment on what kind of real world inputs display such skew.  I've updated my score as the responses have removed some of my uncertainties.

---

> > > ### Author Response · Authors · 2023-08-11
> > > **Thank you for your response**
> > >
> > > Thank you for updating your rating and confirming that some uncertainties have been clarified!
> > >
> > > In many real-world datasets, the frequency vector displays skewness. For example, the Zipf’s law [15] states that in practice, a list of measured values often decays at a polynomial rate when sorted. We will highlight the skewness of real-world datasets more in the revision.

---

> > > > ### Comment · Reviewer_q1hW · 2023-08-13
> > > >
> > > > Indeed, I had in mind Zipfian distributions.  My belief is that it is quite common to find data that fits a Zipfian with skewness parameter (polynomial exponent) in the range 1-2, but 3 or higher (which is shown Fig 2b and 2c) is more extreme.  Note also that a distribution with a parameter of 5 means that  after the 10th most frequent item, the subsequent items are at most a 10^{5} factor less frequent than the most frequent one (approximately).  This is such a sharp drop off that we are unlikely to see very many examples outside the top 10 for moderate (hundreds of thousands of examples) input sizes.  The point being, this (skewness parameter 5) may not be the most representative scenario to highlight in the experimental section.

---

> > > > > ### Author Response · Authors · 2023-08-16
> > > > > **Thank you for your response**
> > > > >
> > > > > Thank you for your engaged discussion with us.
> > > > >
> > > > > Zipf’s distributions are one set of exemplar distributions that possess skewness. There are many other examples, e.g., a distribution with $k$ high-density indexes and $d-k$ indexes where the density decays polynomially. Note that in this case, our two-approach method needs slight revisions and should not be directly applied.
> > > > >
> > > > > We would also like to highlight that while Hybrid Sketch is (significantly) better than Fresh/Shared Sketch when the distribution is light-tailed, Hybrid Sketch is no worse than Fresh/Shared Sketch if the distribution is not light-tailed. So for a fixed communication budget, using Hybrid Sketch causes no accuracy loss in any case, and can potentially lead to a big accuracy gain in good cases.

---

> > > > > > ### Comment · Reviewer_q1hW · 2023-08-16
> > > > > >
> > > > > > Thank you for the thoughtful responses.  It's an important point that the hybrid sketch is no worse than the fresh sketch approach, whatever the distribution.  But the paper is more compelling if there can be examples shown of real-world frequency distributions that elicit a clear separation, and which can be argued represent a common case.

---

> > > > > > > ### Author Response · Authors · 2023-08-17
> > > > > > > **Thank you for your response**
> > > > > > >
> > > > > > > We agree with your comments. As shown in Figure 3, the C4 dataset is one such real-world dataset where Hybrid Sketch clearly separates from Fresh/Shared Sketch.

---

### Official Review · Reviewer_vZGP · 2023-07-03

**Soundness:** 3 good
**Presentation:** 2 fair
**Contribution:** 3 good
**Rating:** 6
**Confidence:** 3

**Summary:**

The paper considers the problem of federated frequency estimation under the Secure Summation constraint. Motivated by the tail-bound analysis of CountSketch, the authors propose a two-round communication approach that can significantly reduce the sketch size and improve the overall communication size. Later, the authors extend the algorithm to a multi-round HybridSketch algorithm and improve the protocol's scalability. This paper also carefully analyzed their proposed method's trade-offs between accuracy and privacy. The proposed algorithm shows performance improvements in real-world datasets (Gowalla and Colossal Clean Crawled Corpus).

**Strengths:**

Federated frequency estimation is a very important task in federated analytic and has many applications. The main idea is intuitive. Many real-world data distributions exhibit skewness, while the chen et al. showcases the lower bound for this task is $nlogd$ bits of communication per client, this work takes the distribution into account to reduce the communication costs.

**Weaknesses:**

The experiments showcase good estimation accuracy improvement. One may wonder how the communication latency changes between the baselines and hybrid sketch.

**Questions:**

What is the difference in communication latency between the proposed algorithm vs. baselines?

Use different ticks or line styles to better distinguish between the lines in figures.


**Limitations:**

No limitations

---

> ### Author Rebuttal · Authors · 2023-08-08
>
> We appreciate your positive evaluation! Please find our answers to your specific questions below.
>
> ---
> **Q1**. “What is the difference in communication latency between the proposed algorithm vs. baselines?”
>
> **A1**. In our experiment, the client/server communication is simulated so the latency is ignorable. In practice, we expect that the communication latency is proportional to the sketch size (which determines the message length). So the difference between our proposed method vs. baselines in the sketch size vs. error plots (see Figure 1(b)(c)(e)(f), Figure 2, and Figure 3) reflects their communication latency difference in practical applications. For example, in Figure 3, the dashed gray lines show that for a fixed error, the hybrid sketch size is much smaller than the fresh and shared design.
>
> ---
> **Q2**. “Use different ticks or line styles to better distinguish between the lines in figures.”
>
> **A2**. We will revise the plots as you suggested.

---

### Official Review · Reviewer_gbH4 · 2023-07-06

**Soundness:** 3 good
**Presentation:** 4 excellent
**Contribution:** 3 good
**Rating:** 7
**Confidence:** 3

**Summary:**

This paper tackles the problem of Federated Frequency Estimation (FFE) by leveraging the Federated CountSketch algorithm (Algorithm 2 in this paper). Compared to previous FFE methods the authors argue that if the tail of the underlying frequencies is light or small (\emph{i.e., } the smaller frequencies have considerably smaller values). Furthermore, such a lighter tail provides two benefits, (1) The FFE error decreases and (2) A smaller width can be afforded in the Federated CountSketch algorithm that leads to a direct reduction in the client communication costs. Under a Federated setting, both advantages are meaningful. In the first half of the paper, the authors provide ways to leverage this lighter tail in the Federated setting. While in the second half, the authors provide FFE algorithms that work for a large number of clients by running CountSketch over multiple groups of clients. Finally, the paper also provides a Differential Privacy (DP) guarantee and estimation error while running the FFE algorithm over multiple rounds.

Overall, the authors provide a clear and unique analysis that furthers the FFE literature by reduction of the estimation error and sketch width when leveraging the (almost optimal) CountSketch algorithm. The empirical evaluations provide a complete picture of the impact of the suggested method. Minor comments regarding the two-phase method, empirical evaluations, and DP section have been provided below.


**Strengths:**

The paper tackles several challenges in the FFE literature. Primarily the authors demonstrate FFE algorithms for the cases when (1) the number of clients is low to moderate and (2) for a high number of clients. They also tackle the case when Differential Privacy (DP) is introduced in FFE deployments.

1. The core idea that the authors leverage is the reduction in the estimation error while using the CountSketch algorithm when the underlying frequencies have a lighter tail. The authors argue that when the data is distributed in such a (lighter-tail) fashion, the previous estimation error bounds are unable to leverage these tighter bounds to provide real-world benefits. Using Corollaries 2.2-2.4 the authors demonstrate a reduction in the estimation error and the sketch width and thus the communication cost of the clients.
2. Furthermore, the paper suggests that when the clients are high in number, simply computing CountSketch over a single round might not be feasible. They demonstrate that the naive application of CountSketch in the multi-round setup is not enough. Thus, they suggest a new algorithm “Hybrid Sketch, " extending CountSketch for the multi-round setup. The Hybrid Sketch algorithm by maintaining the same hash buckets can combine the analysis of frequency estimation across rounds rather than analyze each separately.
3. Finally, a DP alternative of Hybrid Sketch is provided that follows similarly to previous approaches.


**Weaknesses:**

1. For the two-phase method in section 2, the authors suggest that frequencies follow the Zipf law; thus, they estimate the polynomial coefficients by a kind of pilot study on a small number of clients. It is however unclear how the instance optimal method can leverage Eqn. (2) with the unknown frequencies to estimate the sketch width. Is the two-phase method employed for the instance optimal method as well? Please consider adding relevant clarification for all three methods.
2. For the DP analysis consider mentioning the exact mechanism used to add noise. At first glance, it might not be clear if the method employs local DP or central DP.
3. In certain cases, the idea that the frequencies have a lighter tail might not be verifiable (before running the sketching algorithm). Such an instance may occur if the communication or DP budget is constrained or if there are not enough clients to handle both the pilot and the actual sketching algorithm under DP constraints. Thus, adding this limitation clearly toward the end of the article will add clarity.



**Questions:**

The paper is well-written and conveys interesting results for performing Federated Frequency estimation over various scenarios. I do not have major concerns regarding the paper. A few minor suggestions have been pointed out in the weaknesses section.



**Limitations:**

The primary limitation of the proposed method has been provided in point 3 of the weaknesses section. The authors can consider including a limitations/discussion section that highlights such challenges.

---

> ### Author Rebuttal · Authors · 2023-08-08
>
> Thank you for supporting our paper! We will address your comments as follows.
>
> ---
> **Q1**. “For the two-phase method in section 2, the authors suggest that frequencies follow the Zipf law; thus, they estimate the polynomial coefficients by a kind of pilot study on a small number of clients. It is however unclear how the instance optimal method can leverage Eqn. (2) with the unknown frequencies to estimate the sketch width. Is the two-phase method employed for the instance optimal method as well? Please consider adding relevant clarification for all three methods.”
>
> **A1**. In the instance optimal method, the sketch size is computed based on eq. (2) by an oracle. The instance optimal method is for demonstrating the correctness of our theoretical understanding and is not a practical method. We will provide detailed clarifications for all three methods in the revision.
>
> ---
> **Q2**. “For the DP analysis consider mentioning the exact mechanism used to add noise. At first glance, it might not be clear if the method employs local DP or central DP.”
>
> **A2**. We employ central DP. We will clarify this in the revision.
>
> ---
> **Q3**. “In certain cases, the idea that the frequencies have a lighter tail might not be verifiable (before running the sketching algorithm). Such an instance may occur if the communication or DP budget is constrained or if there are not enough clients to handle both the pilot and the actual sketching algorithm under DP constraints. Thus, adding this limitation clearly toward the end of the article will add clarity.”
>
> **A3**. We agree with this limitation and will clarify this in the revision.

---

> > ### Comment · Reviewer_gbH4 · 2023-08-21
> >
> > Thank you for taking the time to address my concerns. I believe the paper adequately addresses all of my concerns including the additional minor ones. Therefore, I will keep the existing rating of the paper unchanged.

---

### Official Review · Reviewer_2XAM · 2023-07-08

**Soundness:** 3 good
**Presentation:** 3 good
**Contribution:** 2 fair
**Rating:** 5
**Confidence:** 3

**Summary:**

This paper explores several variants of the count sketch method for federated frequency estimation (FFE).
- With only one communication, they provide a refined instance-dependent analysis for CountSketch and find that the sketch size depends on unknown problem-dependent quantities.
They then propose a two-phase approach to first learn these quantities first and then perform FFE.
- They also consider the case where multiple communication is available, they explore several variants of count sketch methods to utilize the multiple communication rounds, provide theoretical analysis, and conduct numerical experiments to verify their findings.
- They also explore a differential private extension.

======= after rebuttal =======

I have read the author's rebuttal which addresses most of my concerns.

The remaining issue from my perspective is the lack of an instance-dependent lower bound which is left as a future work, as claimed in the rebuttal.

Hence, I tend to increase the point.

**Strengths:**

They conduct a systematic investigation of FFE problems and provide theoretical analysis.

The paper is well-written and easy to follow.

**Weaknesses:**

The paper tries to study a lot of stuff, which, however, left many questions untouched. See the Questions part.

There seems not much novelty in the theoretical analysis.

The benefit of instance dependence on the estimation guarantee is not well illustrated.

Though the author provides a lot of numerical experiments to validate their theoretical predictions, it is still unclear how the proposed method would affect practical training where (overparameterized) neural networks are used.

**Questions:**

1. From Figure 1 (a)(d), I find that given a target $\ell_{\infty}$ error, the minimax optimal curve often achieves a smaller $\ell_{\infty}$ error than the instance optimal one, though the latter aligns with the straight line $y=x$ better. It seems that the minimax optimal one is better than the instance optimal one in the achieved accuracy, which is quite counterintuitive. Is there anything I missed? Could the author provide some explanation?
2. Could the author provide some intuitive explanation about why Hybrid Sketch could be better than Fresh Sketch? Note that Hybrid Sketch shares a set of bucket hashes but uses independent sets of sign hashes. Why this dependence on shared bucket hashes is better than independent bucket hashes? What about independent bucket hashes and shared sign hashes?
3. Is it possible to provide any lower bound to show the instance-dependent upper bounds are tight?
4. Line 316, when $n$ is fixed, why increasing $M$ improves the estimation error? In Theorem 4.1, the estimation error $\sqrt{\frac{\sum_{i>W}\left(F_i^*\right)^2}{W}}$ seems to have nothing to do with $M$ in the worst case.
5. Is it possible to extend the methods developed in this paper to empirical risk minimization where gradients of nonconvex models (such as neural networks) are used?

**Limitations:**

N/A.

---

> ### Author Rebuttal · Authors · 2023-08-08
>
> Thank you for your comments and suggestions. We address your concerns as follows.
>
> ---
> **Q1**. “From Figure 1 (a)(d), … It seems that the minimax optimal one is better than the instance optimal one in the achieved accuracy, which is quite counterintuitive. Is there anything I missed? Could the author provide some explanation?”
>
> **A1**. We respectfully point out that Figures 1(a) and (d) imply that the minimax optimal curve is **worse** than the instance optimal curve in terms of computing a minimal necessary sketch size. Observe that the minimax optimal curve archives a smaller error than the instance optimal curve, while both are smaller than the targeted error. Note that Count Sketch with a larger sketch size archives a smaller error. So Figures 1(a) and (d) show that the minimax optimal method (eq. (3)) computes an unnecessarily large sketch size to achieve a targeted error, causing a waste of bandwidth, while the instance optimal method (eq. (2)) computes a more accurate sketch size. This is also illustrated in Figures 1(b) and (e).
>
> ---
> **Q2**. “Could the author provide some intuitive explanation about why Hybrid Sketch could be better than Fresh Sketch? Note that Hybrid Sketch shares a set of bucket hashes but uses independent sets of sign hashes. Why this dependence on shared bucket hashes is better than independent bucket hashes? What about independent bucket hashes and shared sign hashes?”
>
> **A2**. Fresh Sketch computes an “average of medians”, that is, produces well-concentrated estimates of each local frequency independently and then takes the average to produce an estimate of the global frequency. In comparison, Hybrid Sketch computes a “median of averages”, that is, 1) produces high-variance estimators for each local frequency, then 2) averages them to obtain low-variance estimators for global frequency, and finally 3) uses the median trick to amplify success probability.
>
> Fresh Sketch is less effective than Hybrid Sketch because the average of independent well-concentrated estimates is less well-concentrated compared to the median of independent low-variance estimates. This is because, for a good event (where the error is small) to happen in the former method, multiple independent good events have to happen simultaneously, which leads to an invocation of the union bound. In contrast, the good event in the latter method only fails with at most exponentially small probability.
>
> In Step 2) of Hybrid Sketch, we use fresh sign hashes and shared bucket hashes to reduce estimation variance. The fresh sign hashes allow cancellation of errors when averaging the (high-variance) local estimators, which leads to an error bound that depends on the heterogeneity vector rather than the global frequency vector. The shared bucket hashes allow the error cancellation to happen in the same bucket, avoiding interface from other buckets, which leads to an error bound that depends on only the tail of the heterogeneity vector rather than the entire vector.
>
> ---
> **Q3**. “Is it possible to provide any lower bound to show the instance-dependent upper bounds are tight?”
>
> **A3**. Minimax lower bound exists in the literature that justifies the sharpness of the upper bound in the worst case, see, e.g., Theorem 8.1 in [13]. However, whether or not there is an instance-dependent lower bound that matches the best-known instance-dependent upper bound is still an open problem to the best of our knowledge, and we will leave that for future work. Our conjecture is that the current instance-dependent upper bound (e.g., Proposition 2.1) can be improved.
>
> ---
> **Q4**. “Line 316, when $n$ is fixed, why increasing $M$ improves the estimation error? In Theorem 4.1, the estimation error $ \sqrt{ \frac{\sum_{i>M} (F_i^*)^2} {M} } $ seems to have nothing to do with $M$ in the worst case.”
>
> **A4**. Thank you for pointing out this typo. We will revise line 316 to “... a larger number of rounds M improves the estimation error **in non-worst cases, e.g., when the local frequency vectors are nearly homogenous**...”
>
> ---
> **Q5**. “Is it possible to extend the methods developed in this paper to empirical risk minimization where gradients of nonconvex models (such as neural networks) are used?”
>
> **A5**. Good question. Our focus in this work is a federated frequency estimation problem. However, our analysis for Count Sketch and Hybrid Sketch can be applied to general vector recovery problems as well. Therefore, we expect our methods can be applied to estimate the gradient vectors in the ERM problem. We will comment on this in the revision.
>
> ---
> **Q6**. “There seems not much novelty in the theoretical analysis.”
>
> **A6**. We use some standard techniques to derive our bounds. However, we’d like to emphasize that both the considered multi-round FFE problem settings and results are new. It also requires non-trivial efforts to establish new effective algorithms such as Hybrid Sketch, to allow the sketching method to adapt to the hardness of the instance, and to make the algorithm differentially private.
>
> ---
> **Q7**. “The benefit of instance dependence on the estimation guarantee is not well illustrated.”
>
> **A7**. We hope our explanations about Figure 1 in **A1** have clarified the benefits of instance-dependent methods over the minimax optimal method. We are happy to improve our presentation further if you have specific suggestions!
>
> ---
> **Q8**. Though the author provides a lot of numerical experiments to validate their theoretical predictions, it is still unclear how the proposed method would affect practical training where (overparameterized) neural networks are used.
>
> **A8**. We’d like to emphasize that our focus in this work is the federated frequency estimation problem and neural network training is beyond the scope of this work. We agree that extending our methods to neural network training is an interesting future direction.

---

### Decision · Program_Chairs · 2023-09-21

**Decision:**

Accept (poster)

**Comment:**

All reviewers felt that this paper made a solid contribution to the growing body of literature on private frequency estimation, even though the novelty of the theoretical analysis was perhaps a bit limited. The paper was well written and should be of interest to the broad NeurIPS community.